# Clinical, Virological Characteristics, and Outcomes of Treatment with Sofosbuvir/Ledipasvir in Two Pediatric Patients Infected by HCV Genotype 4

**DOI:** 10.3390/cells8050416

**Published:** 2019-05-05

**Authors:** Nadia Marascio, Maria Mazzitelli, Grazia Pavia, Aida Giancotti, Giorgio Settimo Barreca, Chiara Costa, Vincenzo Pisani, Giuseppe Greco, Francesca Serapide, Enrico Maria Trecarichi, Francesco Casalinuovo, Maria Carla Liberto, Giovanni Matera, Carlo Torti

**Affiliations:** 1Department of Health Sciences, Unit of Clinical Microbiology, “Magna Graecia” University, 88100 Catanzaro, Italy; nadiamarascio@gmail.com (N.M.); grazia_pavia@libero.it (G.P.); giancotti@unicz.it (A.G.); gbarreca@unicz.it (G.S.B.); mliberto@unicz.it (M.C.L.); gm4106@gmail.com (G.M.); 2Department of Medical and Surgical Sciences, Unit of Infectious and Tropical Diseases, “Magna Graecia” University, 88100 Catanzaro, Italy; m.mazzitelli88@gmail.com (M.M.); c.costa@materdominiaou.it (C.C.); v.pisani@materdominiaou.it (V.P.); grecopep@gmail.com (G.G.); fraserapide@gmail.com (F.S.); em.trecarichi@unicz.it (E.M.T.); 3Institute for Experimental Veterinary Medicine of Southern Italy, 88100 Catanzaro, Italy; francesco.casalinuovo@cert.izsmportici.it

**Keywords:** pediatric patients, Hepatitis C virus 4, sofosbuvir/ledipasvir

## Abstract

Direct-acting antiviral drugs to cure infections with Hepatitis C virus (HCV) achieve a sustained virological response (SVR) in more than 90% of adult patients. At present, clinical trials are ongoing and real-life data are still limited in children. Herein, we report two cases of pediatric patients treated with fixed-dose combination of sofosbuvir/ledipasvir, already approved to treat HCV4 genotype. Both young girls achieved SVR even though HCV4 isolates carried L28M and M31L NS5A resistance-associated substitutions (RASs). Therefore, possible effects of these RASs merit further study, especially in children.

## 1. Introduction

Vertical transmission of Hepatitis C virus (HCV) occurs in 3–10% of pregnant women worldwide, becoming the most common route of transmission in children since 1992. Among children who acquire the infection vertically, only 25% spontaneously clear the virus [1,2,3]. Therefore, 54–86% children develop chronic infections [4]. Mechanisms of mother-to-child transmission (MTCT) are not well understood, but it is known to occur from the first trimester. In 2018, the American Association for the Study of Liver Diseases (AASLD) and the Infection Diseases Society of America (IDSA) recommended universal screening among pregnant women in the United States [5].

In 2017, the Food and Drug Administration (FDA) and the European Medicines Agency (EMA) approved sofosbuvir (SOF)/ledipasvir (LDV) for pediatric patients who were chronically infected by HCV [6]. Prescription criteria are based on age (children older than 12 years) and weight (children exceeding 35 kg) [6]. Clinical trials in younger children are ongoing; however, preliminary data showed a good safety and efficacy of SOF/LDV in adolescents with an SVR rate of 98% [7]. Pangenotypic regimens are currently under study in children but are not yet approved by FDA [5]. However, in Italy, we faced the paradoxical situation that SOF/LDV was validated to treat adolescents infected by HCV1 or HCV4, but drugs were not available due to lack of indication set by the Italian Medicinal Agency (Agenzia Italiana del Farmaco, AIFA), subsequently amended. 

Despite high rates of SVR to direct-acting antiviral agents (DAAs), treatment failure occurs in 5% of patients chronically infected by HCV [8]. Failure is frequently associated with pre-existing or selected resistance-associated substitutions (RASs) [9]. Population sequencing is currently used to detect RASs, with a 15% cutoff [8]. Particularly, NS5A RASs could affect treatment success and persist for years after treatment failure [9,10]. RASs clearly related to treatment failure are reported only in adult people [11]. The aim of the study was to explore correlations between nonsynonymous substitutions and therapy response in the two pediatric patients infected by HCV4 and treated with SOF/LDV in our center. 

## 2. Materials and Methods

### 2.1. Ethic Statement

The study was approved by the Ethical Committee of the “Mater Domini” University Hospital of Catanzaro, Italy. Written informed consent was obtained from each patient in accordance with the principles of the Helsinki Declaration (World Medical Association General Assembly, Seoul, Korea, 59 October 2008).

### 2.2. Clinical Data 

The two pediatric patients were naïve to any prior HCV treatment. They started SOF/LDV (400 mg/90 mg once daily) in January 2018 for 12 weeks. Both reached SVR and are still on follow-up without reporting any side effect.

#### 2.2.1. Case Report 1

A 13-year-old Italian female patient (Patient 1) was infected from the mother at birth. She has been in follow-up at our outpatient clinic from 2014. She was diagnosed to be infected by HCV in 2007, at the age of three years. For this reason, she was admitted to another hospital and was discharged with diagnosis of hepatic steatosis, obesity, and chronic hepatitis by HCV. At baseline, she presented an infection with HCV genotype 4. Interferon-based treatment has not been prescribed for toxicity constrains. From 2012 to 2017, a rapid progression of liver fibrosis at liver elastometry was observed (liver stiffness worsened from 4KPa in 2012 to 8KPa in 2017), so we decided to treat her with DAAs. 

#### 2.2.2. Case Report 2

A 16-year-old Syrian female patient (Patient 2) who arrived in Italy in 2015. She was born from a positive HCV mother and received several blood transfusions for severe anemia. She came to observation in 2015. She was also affected by cerebral palsy, cryoglobulinemia, skin lesions at her hand and feet, and moderate fibrosis at transient elastography (10.1 KPa). Also for this patient, DAAs treatment was indicated. 

### 2.3. Diagnostic Procedures

HCV–RNA viral load was determined using Cobas AmpliPrep/Cobas TaqMan HCV quantitative test v2.0 (Roche Diagnostics, Milan, Italy) with a quantification range of 15 to 100 million IU/mL. Subtyping was performed by Versant HCV genotype v2.0 assay (LiPA) (Siemens, Healthcare Diagnostic Inc., Tarrytown, NY, USA). Fibrosis stage was estimated by transient elastometry (FibroScan^®^, Echosens, Paris, France), interpreted as in References [10,12,13], and abdomen ultrasound was performed at baseline and after the end of treatment. 

### 2.4. Population Sequencing

Viral RNA was extracted from 500 µL serum using the NUCLISENS^®^ easyMAG^®^ (bioMérieux, Florence, Italy). Serum samples taken from healthy subjects were used as negative controls. RNA was reverse-transcribed by the High-Capacity cDNA Reverse Transcription Kits protocol (Applied Biosystems, Foster City, CA, USA). The NS5A and NS5B regions were amplified as previous reported [11], using nested PCR by MasterMix 2.5x, 5prime (Quantabio©, Hamburg, Germany). The PCR products were purified by PCR Illustra MicroSpin S-300 HR Columns (Gelifesciences, Buckinghamshire, UK) in accordance with the manufacturer’s instructions. Sequencing reactions were performed by means the traditional dideoxy chain termination method (ABI PRISM 3500 genetic analyzer, Applied Biosystems). Sequences were aligned with the MUSCLE algorithm [14] and manually edited by MEGA v.7 [15]. The newly generated NS5A and NS5B sequences can be retrieved from GenBank^®^ under accession numbers: MK814770-MK814771-MK814772-MK814773.

### 2.5. Subtyping and Phylogenetic Analyses

To confirm subtype, newly generated NS5A and NS5B sequences were submitted to the Oxford HCV Automated Subtyping Tool v2.0 [16] and the COMET HCV typing tool [17]. The NS5B phylogenetic tree was constructed by the generalized time reversible (GTR) nucleotide substitution model with gamma (Γ)-distribution by PhyML v3.0 online version [18], incorporating reference sequences available from the Los Alamos National Laboratory (LANL) HCV Sequence Database [19]. A phylogenetic tree was visualized with FigTree v1.4.2 (Institute of Evolutionary Biology, University of Edinburgh, Edinburgh, Scotland). 

### 2.6. Genetic Variability Analysis

Polymorphisms and RAS on NS5A and NS5B target regions of DAA–IFN free regimen using Geno2pheno[hcv]0.92 tool [20] and recent literature data [21] were interpreted. Nonsynonymous substitutions at positions 282, 316, 368, 411, 414, 448, 553, 554, 556 on polymerase region were not evaluated.

## 3. Results

Both patients underwent treatment with SOF/LDV for 12 weeks without reporting any side effects. SVR was demonstrated after three months from the end of treatment. Improvement of alpha-fetoprotein and liver stiffness was observed. Table 1 reports hematobiochemical and laboratory parameters of the two patients before treatment and along the follow-up.

Viral isolates from both patients were classified as HCV4 by Versant HCV genotype V2.0 assay: The isolate from Italian patient was classified as subtype 4d, while the isolate from Syrian patient as subtype 4a by both Oxford and COMET subtyping tools and confirmed by phylogenetic analysis of NS5B region (Figure 1). 

To explore whether this response was eventually achieved notwithstanding presence of RASs in dominant viral population (15% of viral *quasispecies*), we searched for nonsynonymous substitutions on NS5A and NS5B target regions at baseline. The HCV4d isolate carried baseline M31L LDV RAS (Figure 2a) plus V34I, F36L, R41K, T58P, D105N, D126E polymorphisms, while they harbored N62S, I116V, R127Q, T130N, and D189S polymorphisms on the NS5B region. The HCV4a isolate carried L28M LDV RAS (Figure 2b) plus K44R, V53M, K56T, I99V, and V130I polymorphisms in the NS5A region; by contrast, the NS5B region harbored several polymorphisms, but they are not related to resistance yet.

## 4. Discussion

Since an effective treatment is now available, it is important to diagnose and treat all children who are chronically infected by HCV. First, for adolescent patients and children, the SOF/LDV combination allows to achieve SVR rates of 98% [7,22]. Moreover, an ongoing Phase II, multicenter open-label trial (NCT 02249182) reported no serious adverse events in HCV1-infected patients during treatment [23]. 

Determination of correct viral subtype is still crucial to prescribe tailored therapy and to foresee treatment response when non pangenotypic regimens are prescribed [13,21]. Reverse hybridization line probe assay (LiPA) misclassifies around 11% of analyzed samples, and it is not able to differentiate among HCV4a/c/d subtypes [24]. In our cases, the NS5B coding region was sequenced by population sequencing for subtyping purposes. HCV subtyping mirrored infection by country of origin. Indeed, the Syrian patient came from an area where subtype 4a dominates endemic infection [25]. By contrast, the Italian female came from an area of the Calabria Region where subtype 4d has a very high prevalence [26,27]. 

The dominant viral population isolated from our patients carried L28M and M31L RASs on HCV4a and HCV4d NS5A target regions, respectively. These RASs were detected in adult patients infected by HCV4 who failed LDV [21], but their actual impact on clinical response is unclear, especially for HCV4. Indeed, definitive demonstrations of a clinical impact would require evidence that prevalence of a mutation at baseline is significantly higher in patients failing a specific treatment than in those with treatment success. Demonstrations are lacking especially for HCV4, because studies about possible correlations between RASs and treatment failure included very few patients infected by this genotype. In 2016, Abergel et al. [28] reported 3/44 patients who did not achieve SVR after SOF/LDV, among whom only 1 had L28M RAS pretreatment and post-treatment. By contrast, all 44 patients had naturally occurring L31M RAS [28]. Analysis of resistance by Dietz and colleagues on 18 NS5A sequences showed L28M in 39% and L31M/V in 17% of viruses from patients who failed [29]. Moreover, phenotypic resistance associated with these mutations has not been demonstrated yet. Importantly, such data are completely lacking in children infected by HCV4. Indeed, both children achieved SVR notwithstanding L28M and M31L RASs. Clearly, our data are preliminary, both because these two are simple case reports and because we did not sequence the entire target gene implicated in SOF resistance, but the only, potentially most important mutation related to SOF resistance in HCV4 that we did not assess was S282T [20]. This mutation was indicated to confer reduced HCV susceptibility to SOF by a single study [30], but its clinical impact has not been demonstrated yet. Indeed, the emergence of S282T was rare in patients who failed SOF-based regimens, and retreatment with SOF-based regimens appeared to be successful in patients who failed, displaying S282T mutation in the viral population [31]. Thus, even though this mutation had been present, we would not have been able to exclude that the mutations detected in our study, which are believed to confer resistance to LDV, were not clinically relevant. In fact, since SOF was co-administered, SVR could have been achieved in our patients because SOF activity compensated for LDV resistance (either assuming that S282T was present or not). Notwithstanding this limitation, we feel that our report is significant from a strategic point of view because it may suggest that the currently available combinations of two drugs, including SOF (in particular SOF + LDV from our study), may still work when resistance mutations are present. 

Further, in addition to L28M and M31L, we found a wide range of substitutions whose impact is not evident from our study, which have been already studied in adults but whose significance in children was not explored before. HCV4d isolated from the first of our patients carried also V34I, F36L, R41K, T58P, D105N, and D126E polymorphisms, while HCV4a from the second patient additionally harbored K44R, V53M, K56T, I99V, and V130I polymorphisms on the NS5A coding region [32]. It has already been reported that simultaneous polymorphisms could modify affinity between drug and NS5A phosphoprotein [32]. In 2018, Minosse et al. [33] observed T58P substitution in 69.2% of cases among 50 HCV4 positive patients; this substitution has already been reported in two patients failing SOF/LDV therapy [34]. Other polymorphisms, such as V34I, V130I, and D105N, were found in 99.1%, 99.6%, and 64,3% of viral population, respectively, using next-generation sequencing at SVR12 [35]. Basically, several substitutions, such as L28M (10.7%) and M31L (16.1%) RASs and T58P (85.7%) polymorphism detected in a significant proportion of HCV4 strains were found [35]. More studies are necessary to explore whether the presence of these variants correlates with treatment failure in children. To the best of our knowledge, the ongoing clinical trials including adolescent aged 12–17 years are evaluating efficacy and safety of treatment, including adverse events [7]. 

Lastly, since the immune system can influence HCV control and intrahost selection of viral variants [36,37], treatment response and HCV evolution merit investigation in children, whose immune system response to the virus may differ compared to adults. Different virus–host interactions in children may also have a different impact on the influence of specific RASs on clinical response to DAA treatment in children compared to adults. Recently, possible markers in the induction of chronic infection in children (IL-17 production and the phenotype of lymphocytes subsets) have been proposed [38], but many other several cells and related biomarkers are involved in HCV infection progression [39]. A larger number of patients need to be enrolled, in order to study the interaction between HCV and pediatric immune system.

In conclusion, these case reports underline the importance of treating children affected by HCV, which is now feasible even in conditions where achievement of SVR is deemed to be impaired by pre-existing RASs. The impact and mechanisms of RASs impairing clinical response in children merits further elucidation with studies conducted in this specific population of patients.

## Figures and Tables

**Figure 1 cells-08-00416-f001:**
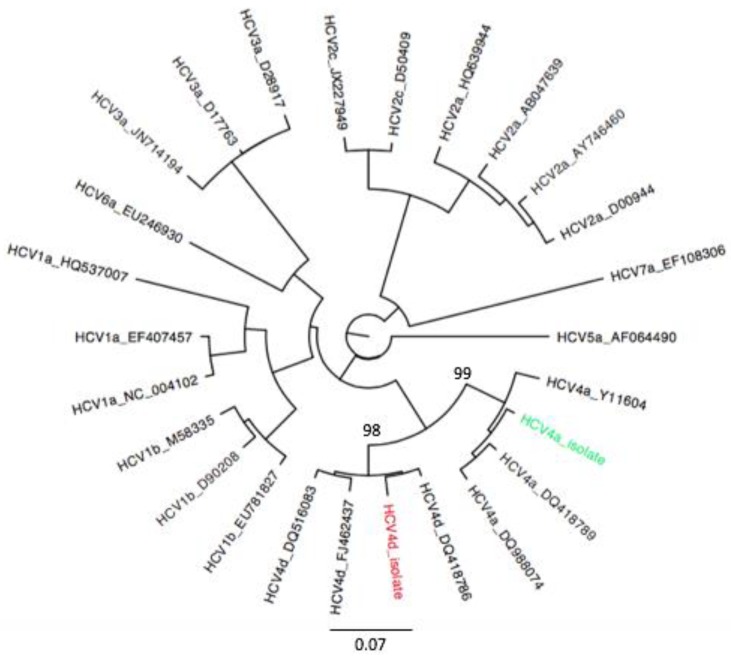
PhyML phylogenetic trees was estimated using 24 Hepatitis C virus (HCV) reference sequences (black), HCV4d (red) and HCV4a (green) sequences isolated in this study. The cluster reliability was supported by bootstrapping with 100 replications. Bootstrap support values are shown for the HCV4d/4a clades. The scale bars at the bottom of the figure represent genetic distance.

**Figure 2 cells-08-00416-f002:**
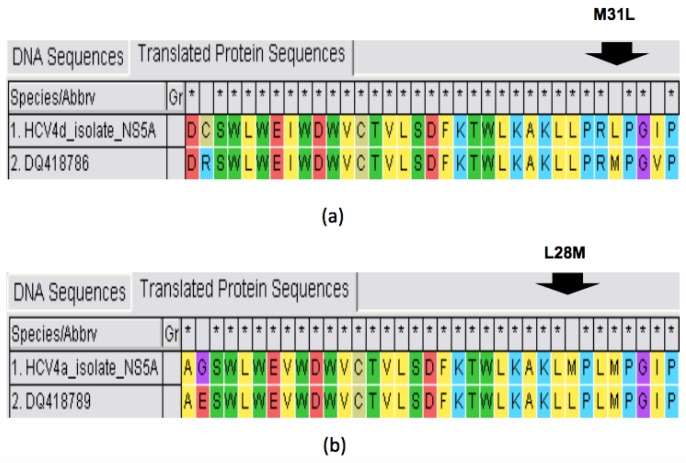
RASs in the HCV4d (**a**) and HCV4a (**b**) NS5A target region vs. subtype reference sequence visualized by MEGAv7 software.

**Table 1 cells-08-00416-t001:** Patient characteristics at baseline and along the follow-up.

	Patient 1				Patient 2			
	Baseline	EOT	SVR12	SVR24	Baseline	EOT	SVR12	SVR24
**Quantitative Variables**								
**Viral load (UI/mL)**	7.490.000	TND	TND	TND	242.000	TND	TND	TND
**White cells (×10^3^/mL)**	8.000	4.780	4.820	4.520	7.010	6.090	8.290	6.080
**Hemoglobin (g/dL)**	15.1	13.2	13.7	14.3	13.1	12.8	12.6	11.7
**Platelet count (×10^3^/mL)**	307	223	239	213	315	269	358	283
**Creatinine (mg/dL)**	0.63	0.58	0.55	0.57	0.52	0.53	0.62	0.52
**AST (UI/L)**	49	22	25	21	19	16	16	16
**ALT (UI/L)**	70	17	19	17	15	9	9	8
**γ GT (UI/L)**	39	8	9	8	10	10	10	11
**Total cholesterol (mg/dL)**	108	118	122	130	141	155	154	156
**Triglycerides (mg/dL)**	64	37	73	51	80	58	76	47
**Total bilirubin (mg/dL)**	1.00	0.87	1.11	0.99	0.33	0.40	0.29	0.27
**Albumin (g/dL)**	5.00	4.5	4.7	4.8	4.7	4.8	5.1	4.5
**Glucose (mg/dL)**	84	82	83	98	90	88	84	89
**Alpha-fetoprotein (ng/mL)**	5.89	NA	NA	3.3	3.2	NA	NA	2
**Qualitative Variables (Yes/No)**								
**HIV coinfection**	No	No	No	No	No	No	No	No
**HBV coinfection**	No	No	No	No	No	No	No	No
**Cryoglobulinemia**	No	NA	NA	NA	Yes	NA	NA	Yes

SVR = sustained virologic response, EOT = end of treatment response, TND = target not detected, NA = not available.

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
