# Peer review of "Clinical, Virological Characteristics, and Outcomes of Treatment with Sofosbuvir/Ledipasvir in Two Pediatric Patients Infected by HCV Genotype 4"

_cells, 2019, doi:10.3390/cells8050416_

Round 1
Reviewer 1 Report
The authors partly modified their manuscript following my previous comments. However, the authors did not sequence the NS5B gene implicated in SOF resistance. If the NS5B of these HCV strains were sensitive to SOF, it is not surprising that these patients achieved SVR.
Specific points
1. The authors should make it clear whether the HCV strains have resistance-associated substitutions (RASs) for SOF. If the HCV strains don't have RASs for SOF, there is no novelty in this manuscript.
2. P. 3, line 10. ‘ accession numbers; in submission. The authors should have already obtained the accession numbers. The accession numbers should be provided.
3. P. 5, lines 32-33. ‘ So, our data suggesting that the reported mutations are not able to impair clinical response to LDV in association with SOF are important. ‘ Why do the authors think like this? This sentence is not agreeable.
Author Response
The authors partly modified their manuscript following my previous comments. However, the authors did not sequence the NS5B gene implicated in SOF resistance. If the NS5B of these HCV strains were sensitive to SOF, it is not surprising that these patients achieved SVR.
We partially agree with your comment. Indeed, we sequenced the NS5B until position 210. So, some substitutions found in patients failing sofosbuvir based regimens may have been missed. However, the impact of these mutations on treatment response is under study, so substantial uncertainty would persist regarding your doubt, even if these substitutions have been detected. Indeed, even when the most important substitution (i.e., 282T) is present, patients may respond well to treatment including sofosbuvir. For this reason, we believe that our results should be interpreted conceptually as a further demonstration that the clinical impact of substitutions needs to be studied more in depth.
Specific points
1. The authors should make it clear whether the HCV strains have resistance-associated substitutions (RASs) for SOF. If the HCV strains don't have RASs for SOF, there is no novelty in this manuscript.
This specific points relates to the general comments raised above. To be more specific in answering to this question, the most important mutation missed, the S282T, was indicated to confer reduced HCV susceptibility to sofosbuvir by a single "in vitro" study for HCV 4 and its clinical impact has not been demonstrated yet. Moreover, the emergence of S282T was rare in patients who failed sofosbuvir and re-treatment with this drug appeared to be successful despite its presence. Therefore, although this point has been appropriately discussed as a limitation, we believe that our report may be significant and provoke further investigations.
2. P. 3, line 10. ‘ accession numbers; in submission. The authors should have already obtained the accession numbers. The accession numbers should be provided.
We provided accession numbers. The sequences are retrieved from GenBank under the following accession numbers: MK814770-MK814771-MK814772-MK814773
3. P. 5, lines 32-33. ‘ So, our data suggesting that the reported mutations are not able to impair clinical response to LDV in association with SOF are important. ‘ Why do the authors think like this? This sentence is not agreeable.
We have deleted this sentence.
Reviewer 2 Report
All the concerns have been properly corrected in the revised maunsicript.
Author Response
Thank you for your positive comments.
Round 2
Reviewer 1 Report
The manuscript has been improved considerably.
This manuscript is a resubmission of an earlier submission. The following is a list of the peer review reports and author responses from that submission.
Round 1
Reviewer 1 Report
The authors describe two cases of pediatric patients infected by HCV genotype 4. These two young girls achieved SVR even though HCV 4 isolates carried L28M and M31L NS5A resistance-associated substitutions (RASs). These patients were treated with sofosbuvir/ledipasvir. If the HCV strains were sensitive to sofosbuvir, it is feasible that the patients reached SVR. Therefore, the authors should discuss more about the NS5B with N62S, I116V, R127Q, T130N, and D189S mutation. If these mutations are not related to resistance, it is feasible that these HCV strains were sensitive to sofosbuvir. However, the authors propose that possible effect of RASs in children may be different compared to adults and this merits to be studied further. I don’t feel that their conclusion is supported by the sequence data.
Reviewer 2 Report
In this manuscript, authors reported two pediatric cases with genotype 4 HCV infection successfully treated by sofosbuvir plus ledipasvir even though being infected with resistant-associated variants (RASs). Since those RASs are generally considered as resistant to DAA therapy in adults, the authors speculate that anti-viral response to HCV might be different between children and adults. Since the response as well as adverse events of DAAs for HCV-infected children are unclear, such reports might have important clinical information.
On the other hand, there are some major flaws.
If the authors insist that anti-viral response to HCV-RAS (L28M and L31M) is superior in children, authors should show the response rate to LDV/SOF therapy in those adults infected with HCV-4 RAS (L28M and L31M). On the other hand, since viral response to LDV/SOF therapy is generally known as favorable even with RAS in genotype-4 infection in adults, it is not strange that favorable response to LDV/SOF therapy was observed in children.
It is not described how the authors selected these two cases. Without this description, readers cannot judge if such favorable response could be unbiasedly observed in children.